# The Loss of Structural Integrity of 3D Chitin Scaffolds from *Aplysina aerophoba* Marine Demosponge after Treatment with LiOH

**DOI:** 10.3390/md21060334

**Published:** 2023-05-30

**Authors:** Izabela Dziedzic, Alona Voronkina, Martyna Pajewska-Szmyt, Martyna Kotula, Anita Kubiak, Heike Meissner, Tomas Duminis, Hermann Ehrlich

**Affiliations:** 1Faculty of Chemistry, Adam Mickiewicz University, Uniwersytetu Poznańskiego 8, 61-614 Poznan, Poland; martyna.kotula97@gmail.com (M.K.); anitakubiak9@gmail.com (A.K.); 2Center for Advanced Technologies, Adam Mickiewicz University, Uniwersytetu Poznańskiego 10, 61-614 Poznan, Poland; mpszmyt@amu.edu.pl (M.P.-S.); tomas.duminis@amu.edu.pl (T.D.); 3Department of Pharmacy, National Pirogov Memorial Medical University, Pirogov Str. 56, 21018 Vinnytsia, Ukraine; algol2808@gmail.com; 4Institute of Electronics and Sensor Materials, TU Bergakademie Freiberg, Gustav Zeuner Str. 3, 09599 Freiberg, Germany; 5Department of Prosthetic Dentistry, Faculty of Medicine, University Hospital Carl Gustav Carus of Technische Universität Dresden, Fetscherstraße 74, 01307 Dresden, Germany; heike.meissner@uniklinikum-dresden.de

**Keywords:** chitin, marine sponges, *Aplysina aerophoba*, dissolution, bromotyrosines

## Abstract

Aminopolysaccharide chitin is one of the main structural biopolymers in sponges that is responsible for the mechanical stability of their unique 3D-structured microfibrous and porous skeletons. Chitin in representatives of exclusively marine Verongiida demosponges exists in the form of biocomposite-based scaffolds chemically bounded with biominerals, lipids, proteins, and bromotyrosines. Treatment with alkalis remains one of the classical approaches to isolate pure chitin from the sponge skeleton. For the first time, we carried out extraction of multilayered, tube-like chitin from skeletons of cultivated *Aplysina aerophoba* demosponge using 1% LiOH solution at 65 °C following sonication. Surprisingly, this approach leads not only to the isolation of chitinous scaffolds but also to their dissolution and the formation of amorphous-like matter. Simultaneously, isofistularin-containing extracts have been obtained. Due to the absence of any changes between the chitin standard derived from arthropods and the sponge-derived chitin treated with LiOH under the same experimental conditions, we suggest that bromotyrosines in *A. aerophoba* sponge represent the target for lithium ion activity with respect to the formation of LiBr. This compound, however, is a well-recognized solubilizing reagent of diverse biopolymers including cellulose and chitosan. We propose a possible dissolution mechanism of this very special kind of sponge chitin.

## 1. Introduction

Chitin is one of the fundamental and most abundant structural polysaccharides in nature. In contrast to cellulose, its distribution and isomorphism belong to uni- and multicellular organisms. This polysaccharide has been identified within various skeletal structures of unicellular and multicellular organisms [1], including fungi [2], protists [3], diatoms [4], sponges [5,6,7,8], molluscs [9,10], arthropods [11,12,13,14,15] and fish [16]. N-acetyl-D-glucosamine units linked by β-(1-4)-glycosidic bonds [17] are the principal molecular fingerprints found in three polymorphic forms of chitin—α, β and γ—which exhibit different arrangement of the polymeric chains [18]. Chitin (and its biocomposites with diverse pigments, proteins, and lipids as well as mineral-based phases [19,20,21]) usually provides the stiffness and rigidification of the corresponding exoskeletal structures [5]. 

Traditionally, chitin is characterized as nontoxic, biocompatible, biodegradable, and physiologically inert [22]. Moreover, its fibers are nonallergenic, antibacterial, deodorizing, and moisture-controlling. On the account of these properties, chitin has found applications in the food industry, biomedicine, tissue engineering, wound-dressing material, and controlled drug release, among others [23,24]. It can also be used for the adsorption of industrial pollutants and in the paper-making process [17,22]. Another highly interesting application of chitin is shell biorefinery, where crustacean shell waste can be utilized to produce organonitrogen chemicals for the production of agrochemicals or pharmaceutical products [25,26,27]. Chitin can be chemically transformed to its deacetylated derivative called chitosan. In contrast to chitin, this industrially-produced biopolymer is soluble in aqueous acidic solutions, which opens up broad possibilities for its applications (see for overview [28]). 

One of the new and recently investigated sources of naturally pre-structured chitin are representatives of demosponges (class in the phylum Porifera). For example, selected demosponges of the order Verongiida possess the unique skeletons in the form of three-dimensional (3D) flat or cylindrical chitin scaffolds [8], which are characterized by their mechanical stability as well as their ability to swell due to capillary effects within their microtubular structures [18]. Together with structural rigidity, this makes chitinous scaffolds remarkable candidates especially for applications in tissue engineering and biomedicine [1,24,29,30,31,32,33]. Chitin-based scaffolds of sponges are both chemically stable and thermostable up to 400 °C; this also makes them commonly used objects for extreme biomimetics [34,35] and the development of novel composites and biomaterials [36,37,38,39,40].

Structural biopolymers (i.e. cellulose, spongin, silk) are mostly insoluble. This represents one of the limiting factors for their practical applications in diverse technologies. Additionally, chitin is insoluble in most common solvents [18]. This fact makes it difficult to develop methods for processing and using this relevant polymer [17]. The problem with chitin solubilization is its high crystallization and strong inter- and intramolecular hydrogen bonds [41]. This polysaccharide can be only dissolved in solvents that destroy the hydrogen bonds (H-bonds). Furthermore, most of these solvents are toxic, mutagenic, or corrosive [42]. In the Table 1, examples of already-reported chitin solvents are represented.

In the first study on chitin solvents [43,44,45], the complex between chitin and LiCl that coordinated with the acetyl group was obtained. The complex was soluble in N-methyl-2-pyrrolidone and dimethylacetamide. For the dissolution of chitin chains, such carboxylic acids as formic, dichloroacetic, and trichloroacetic [43,44,45] have also been used previously.

**Table 1 marinedrugs-21-00334-t001:** Overview of chitin solvents.

Solvent	Advantages	Disadvantages	Reference
2-chloroethanol and mineral acid	Dissolving chitin rapidly at room or mildly elevated temperature	Hydrolytic degradation occurs	[45]
Carboxylic acids (formic, dichloroacetic, trichloroacetic)	Dissolving chitin rapidly, usually at room or mildly elevated temperature	Chitin is degrading slowly; solutions of chitin in formic acid are unstable	[43,44,46]
Concentrated phosphoric acid	Dissolving chitin rapidly at room temperature	Chitin is hydrolyzed after a long time in the acid at room temperature	[47]
Hexafluoroacetone sesquihydrate	The solutions formed may be wet or dry spun into filaments, or cast into films or solid articles	Toxicity	[48]
Hexafluoro-2-propanol	No chitin degradation occurs	Toxicity	[49]
CaCl_2_·2H_2_O-saturated methanol	Clear chitin solution easy to regenerate chitin into diverse forms	Chitin solubility depends on the degree of deacetylation and molecular weight	[50,51]
LiCl/N-methyl-2-pyrrolidone (NMP)	Non-degrading solvent	Toxicity	[43,46,52,53]
LiCl/dimethylacetamide (DMA)	Non-degrading solvent	Not all species of chitin can be dissolved; toxicity	[43,46,52,53]
LiSCN	No hydrolysis	High temperatures required	[46,54]
LiI	No hydrolysis	High temperatures required	[54]
LiCl/DMF	Relatively short time (1 h)	Toxicity	[52,55]
NaOH/crushed ice or freezing	Chitin in alkali is stable with respect to degradation	Hydrolysis occurs	[52,56,57,58,59]
NaOH/urea	Little effect on the chitin structure; retaining the degree of deacetylation	Temperature not higher than −20 °C	[52,60]
KOH/urea	Good chitin solubility (~80%)	Deacetylation occurs (ca. 12.5%); Low temperatures required (−25 °C)	[41]
Deep eutectic solvents	No structural degradation	High temperatures required; depolymerization occurs	[61]
Ionic liquids	Dissolve chitin of all polymorphic forms; green solvents	Elevated temperatures required	[42,62]

Solvents for dissolving chitin are often toxic or can cause hydrolysis, depolymerization, or structural degradation (see Table 1), so the challenge of developing a novel method for dissolving chitin remains important. Usually, chemical isolation of chitin scaffolds from corresponding marine sponges is based on an application of up to 10% concentrated NaOH, or Ba(OH)_2_ (see for overview [8,63]). Any reports on the dissolution of such structures being located in these reagents during more than 12 months have been published to our best knowledge. However, we have recently started experiments by isolating chitin scaffolds from *Aplysina aerophoba* demosponge (Figure 1) cultivated under marine ranching conditions [64] using lithium hydroxide solution (LiOH).

For this purpose, the solution of 1% LiOH was used and the process was conducted with the assistance of corresponding ultrasound treatment. Surprisingly, this simple method results in the dissolution of chitinous scaffolds and obtaining of both chitinous amorphous-like matter and bromotyrosines-containing extract. For the first time, we describe this methodology (see Figure 12) and propose a possible dissolution mechanism of this very special kind of chitin.

## 2. Results

Our primary aim was to isolate 3D chitin scaffolds from cultivated *A. aerophoba* sponge (Figure 2a) using a 1% LiOH solution similar to the classical approach reported by us previously [30,65,66]. However, during the procedure, it was observed that the amount of the obtained scaffolds was smaller than expected and the structural integrity of the scaffolds was significantly decreased (Figure 2b). The reason for this was the dissolution of chitin scaffolds in this solvent under the conditions used in the study (see Figure 12). 

### 2.1. Digital Microscopy

In a digital microscopic image (Figure 3) of the partially dissolved scaffold, the residual, pigmented chitin fibers (Figure 3b, arrows) were seen with the surrounding enigmatic structure. To examine this structure, the solution in which the scaffolds were sonicated was dialyzed and lyophilized. The product is shown in Figure 3a,c,d, and it had the structure of a thin (ca. 1 mm) film. Chitin fibers cannot be observed in these samples.

The process of the dissolution of the chitinous fibers is visualized in Figure 4 as the comparison of the digital microscopy images of the chitinous scaffolds of *A. aerophoba* demosponge obtained under similar experimental conditions after treatment with 10% NaOH (Figure 4a,c,e), with the same partially dissolved in 1% LiOH. We have observed the loss of structural integrity of the tubular structure of the scaffold during its layer-by-layer dissolution in LiOH (Figure 4b,d,f).

### 2.2. Fourier Transformed Infrared Spectroscopy (FTIR)

Infrared spectroscopy remains a well-recognized analytical method for the characterization of polysaccharides, including both chitin and chitosan [18]. Figure 5 shows the FTIR spectra of chitin and chitosan standards in comparison to an LiOH-treated sponge chitin sample. All characteristic bands attributed to chitin can be seen in the spectrum of the LiOH-treated samples. The spectrum of the chitosan standard and the LiOH-treated sponge samples differ significantly, so no deacetylation occurred during the process used in the study. No shift of the bands indicates the chemical stability of the polysaccharide [61]. Since there are no noticeable differences in the FTIR spectra of chitin standard and dissolved chitin. We emphasize that the dissolution of chitin in LiOH does not cause any chemical modification of the biopolymer on a molecular level. 

In the spectra of dissolved chitin, the amide bands attributed to the CONH group vibrational modes appeared at 1627 cm^−1^ (amide I), 1557 cm^−1^ (amide II) and between 1309 and 1203 cm^−1^ (amide III). The presence of the amide I doublet is related to the crystalline structure of chitin. The loss of sharpness in this region indicates the loss in crystallinity after the dissolution process. Four strong bands ascribed to the C–O–C and C–O stretching modes appeared at 1154, 1112, 1067, and 1028 cm^−1^. The vibrational absorption band at 1375 cm^−1^ was interpreted as the methyl group rocking. These bands were identical to or very close to the reference spectrum of α-chitin [6,18,67,68,69]. The assignment of the bands is presented in the Table 2. 

### 2.3. X-ray Diffraction (XRD)

The XRD patterns of chitin and chitosan standards and dissolved chitin are shown in Figure 6. The chitosan standard pattern shows two major peaks at 2θ = 11.16° and 20.26° related to the (020) and (200) crystallographic planes, respectively [18]. The XRD pattern of the chitin standard presents characteristic peaks at 9.31°, 12.69°, 19.27°, 20.55°, 23.33° and 26.31° corresponding to the reflections (020), (021), (110), (120), (130) and (013), respectively. After the dissolution process, the peaks at 20.55° and 23.33° disappeared. Dissolved chitin displays the major α-chitin diffraction peaks at 2θ = 9.03° and 19.65° corresponding to the α-type chitin crystallographic plane (020) and (110), respectively. Two minor peaks at 2θ = 12.50° and 26.81° are also visible and can be attributed to the reflections (021) and (013), respectively. These reflections are broad and localized at characteristic positions for α-chitin [18,65]. Combining the XRD results with the SEM images (Figure 6) shows the presence of nanofibers and suggests a nanocrystalline organization of the obtained sample [65]. Broadening of the XRD signals may arise from increased surface areas exposing chains of polysaccharides [65,70]. The peaks in the dissolved chitin spectrum are wider when compared to the reference standard. That may indicate a decrease in the crystallinity of chitin. The presence of a minor peak around 70° and the intensity of the peak associated with the reflection (020) in the dissolved chitin spectrum may be due to impurities. It is expected that the intra- and intermolecular hydrogen bonds are broken in the solution [61]. Since the spectra of the dissolved chitin are similar to those of the reference α-chitin standard and different from the chitosan standard spectrum, no chemical modifications occurred, which validates the FTIR analysis.

### 2.4. Scanning Electron Microscopy (SEM) 

Figure 7 presents SEM images of the sponge chitin fibers isolated after NaOH treatment (a, c) and those after dissolution in LiOH solution (b, d). The difference in the structural integrity is clearly visible. The surface of the NaOH-treated chitin scaffold is rough and monolithic (Figure 7a,c). The micrograph of the dissolved sponge chitin after freeze-drying presents a structure consisting of smooth layers (Figure 7b). A closer look at the apparently disintegrated smooth surface reveals the presence of chitin nanofibers (Figure 7d).

### 2.5. Control Test

In order to conduct the control test with respect to the special activity of LiOH on selected sources of chitin, the samples of snow crab chitin and amorphous chitin standards were sonicated in the lithium hydroxide solution under the same conditions as in experiments with *A. aerophoba* sponge. After 24 days of treatment, we observed that only the chitin isolated from *A. aerophoba* sponge dissolved, while the rest of the chitin samples under the study remained stable (Figure 8). In contrast to the chitin reference standard derived from arthropods, only the chitin isolated from *A. aerophoba* sponge contained bromine in the form of bromotyrosines (for details, see [21,24,63]). The residual amount of Br detected in this partially dissolved chitin by EDX analysis ranged between 0.17 and 0.25 weight percent. Thus, we suggest that the observed phenomenon of this sponge chitin dissolution is based on a possible interaction between Li and Br ions that lead to the formation of LiBr, which is recognized as a well-known reagent for the dissolution of biopolymers, including those based on polysaccharides [71,72,73,74].

### 2.6. CFW Staining and Fluorescence Microscopy for Chitin Identification

Additionally, in order to confirm the presence of chitin in the partially dissolved in 1% LiOH solution *A. aerophoba* skeletal scaffolds, Calcofluor white staining was used. This technique was already used for chitin identification in scaffolds of various species of marine sponges including *A. aerophoba* [64], *A. archeri*, *Ianthella basta* [75], *Ianthella labirintus* [76], etc. The mechanism of the method relies on the compound forming with a strong fluorescence in the blue field of the spectra, while CFW binds the β-glycoside groups of chitin. Both the partially LiOH-dissolved fibers of the sponge chitin scaffold (Figure 9b) and the dried film of the finally-obtained and lyophilized samples (Figure 9d) showed strong blue fluorescence under exceptionally low light exposure time (Figure 9a,c).

### 2.7. Bromotyrosines-Based Extracts

Simultaneously with the isolation of chitin scaffolds, this kind of LiOH treatment leads to the formation of a dark brownish colored, bromotyrosine-containing extract (Figure 10a and Figure 12). After dialysis and drying, it creates a black or brown crystalline-like phase (Figure 10b,c). Figure 10d,e show the FTIR spectra of the obtained *A. aerophoba* extract compared with that of the Isofistularin-3 standard, one of the most well-known bromotyrosines isolated from this verongiid sponge [64,77]. Bands in the range 2840–3500 cm^−1^ are similar on both spectra and show the presence of the hydroxyl (3272 cm^−1^) and CH*_x_* groups (2957, 2924, 2853 cm^−1^). Above 3000 cm^−1^, the band originating from C–H stretching also occurs. The bands at 1639 and 1539 cm^−1^ may be the result of the presence of the amide bond. The peak at 1455 cm^−1^ can be attributed to the aromatic ring stretch. The bands at 1159 and 1037 cm^−1^ may be related to the C–O stretching vibrations. Aliphatic bromo compounds exhibit a band at the frequency between 600–700 cm^−1^, but if several bromine atoms are present, the interpretation is not evident. In the case of brominated aromatic compounds, the presence of the halogen cannot be detected directly [78]. 

Isofistularin-3 standard exhibits peaks characteristic for this compound (3383, 2934, 1660, 1595, 1539, 1450, and 1258 cm^−1^), which are consistent with the literature [79,80,81]. The bands at 3383, 2934, 1660 and 1539 cm^−1^ can be attributed to the hydroxyl groups [81]. 

In the fingerprint region, both the isofistularin-3 standard and the spectra for the extract preparation exhibit very similar peaks. These are the peaks at 1450, 1310, 1258, 1151, 1043, and 739 cm^−1^ in the isofistularin-3 standard spectrum, which correspond well to the peaks in the spectrum of the *A. aerophoba* extract at 1455, 1314, 1252, 1159, 1037 and 737 cm^−1^. The band at 1539 cm^−1^ is located in the same position on both spectra. 

## 3. Discussion

Despite the numerous publications regarding the features of 3D sponge chitin matrices since the first experimental data obtained in 2007 [6], many questions remain open. For example, there is still a lack of information concerning the mechanical properties of the chitinous scaffolds isolated from the sponges. However, these properties are based on structural features of multi-layered, tube-like sponge chitin, where fibrous formations are strongly interconnected within a 3D skeletal construct. Only recently, Machalowski and co-workers [82] presented the following data. The overall macro-scale compressive modulus of chitinous scaffolds isolated from *A. fistularis* marine demosponge that belongs to verongiids has been calculated as ~0.5 kPa. Nothing is known about the possible role of bromotyrosines as cross-linking agents between the chitinous skeletal nanofibers, even though these specific halogenated amino acids are recognized as identification markers for chitin derived from sponges in the Verongiida order [21,63,77]. However, 3,5-dibromotyrosines have previously been reported as cross-linking agents in the cuticle of the Atlantic horseshoe crab (*Limulus polyphemus*) [83], as well as within an operculum scleroprotein from the large whelk *Buccinum undatum* [84]. 

Additionally, the thermal stability of poriferan chitin [85], as well as its resistance to dissolution in alkalis such as NaOH and Ba(OH)_2_ [8,63], remains to be investigated. Therefore, the phenomenon of dissolution of the chitin matrix from a well-studied *A. aerophoba* sponge during its treatment with low concentrated lithium hydroxide, despite its obviousness, led to the emergence of a logical question about the mechanism of the possible effect of this compound on such a specific chitin. 

During the dissolution experiments in this study, it was observed that pigmented, bromotyrosine-containing microfibers of *A. aerophoba* chitin “disappeared”, being transformed into amorphous-like matter (Figure 3b). Consequently, we suggest that lithium hydroxide reacted with the bromotyrosine bromine from the *A. aerophoba* sponge, giving lithium bromide, which finally dissolved the scaffold. A schematic presentation of the process is shown in the Figure 11. The concentration of isofistularin as the main possible source of bromine in this species can reach up to 55 mg per 1 g of dried sponge [64]. It can be suggested that bromotyrosines-based cross-links may be responsible for the interconnectivity of chitinous nanolayers, which form corresponding skeletal microtubes (see Figure 4d,f). None of the already studied alkalis, except for lithium hydroxide here, led to the separation of such tubular structures and their subsequent transformation into membrane-like formations.

Therefore, lithium bromide solution can be a promising candidate for the dissolution of such special chitin as that from *A. aerophoba* demosponge. Moreover, this is confirmed by the fact that other natural polymers that have a limited number of solvents due to the occurrence of hydrogen bonds, such as silk, can be successfully dissolved in lithium bromide, giving an aqueous silk solution. This is due to the fact that LiBr is a chaotropic salt, which can break the inter- and intramolecular H-bonds and dissolve the silk fibers at high concentrations [73]. Feng et al. used lithium bromide solution to dissolve both silk fibroin and cellulose to produce porous sponges with a 3D nanofibrous structure [72]. It was reported that cellulose dissolution in lithium bromide solution causes a decrease in its crystallinity and that it can be dissolved in an aqueous LiBr solution producing cellulose nanofibrils. Upon heating the LiBr solution, the cellulose undergoes complete dissolution [71,86]. The dissolution of chitin in LiBr under diverse, selected conditions has been also reported (for details see [87,88]).

**Figure 11 marinedrugs-21-00334-f011:**
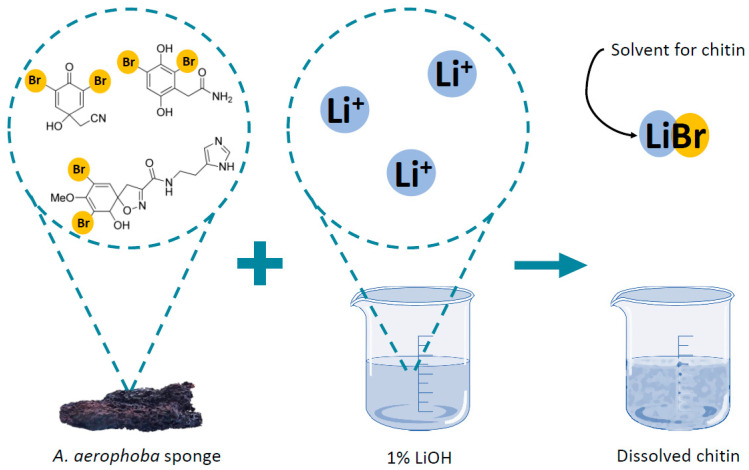
Schematic view of the dissolution mechanism of chitin in *A. aerophoba*. The molecules of selected bromotyrosines have been adapted from [89].

Chitosan can also be dissolved in the aqueous solution of lithium bromide. This process does not induce significant differences in the crystallinity [86].

Traditionally, the main advantage of poriferan chitin is based on its structural integrity on the 3D level and, consequently, the suitability of its application in cell and tissue engineering [31]. The insertion of *A. aerophoba* demosponge (as a renowned source of such ready-to-use chitin [1]) into 1% solution of LiOH at 65 °C leads to the destruction of 3D chitinous constructs; however, it also opens the door for obtaining chitinous film-like structures with nanofibrous architecture (Figure 7d). Despite the loss of the initial tube-like 3D architecture (see Figure 4c,d), sponge chitin still possesses the chemical nature of *alpha*-chitin. No evidence concerning the possible formation of chitosan under conditions used in this study can be shown. Definitively, open questions such as the kinetics of sponge-derived chitin dissolution depending on LiOH concentrations and the temperature of the corresponding reactions will be studied in the near future. In addition, the structural features of LiOH-isolated chitin on the nano-level using TEM and HR-TEM approaches are being planned together with nanoindentation studies.

Intriguingly, the application of LiOH in our case also opens the perspective for optimization of the bromotyrosine isolation method, especially for such a biologically active antitumor drug as isofistularin-3 [77]. Today, the price of 1 g isofistularin-3 is 150,000$ [64]. Due to the cultivation of *A. aerophoba* sponges under marine ranching conditions [90], the isolation of this compound can lead to interesting changes in marine pharmacology and the marine bioeconomy of sponges as well.

## 4. Materials and Methods

### 4.1. Sample Collection

Cultivated *A. aerophoba* sponge was collected in the Adriatic Sea (Kotor Bay, Montenegro) from the marine aquaculture facility from the depths of 3–5 m by scuba diving. Sponge samples were put in Ziploc bags underwater, brought to the laboratory, and washed with water to remove salts. Selected specimens were air-dried for 7 days prior to further treatment and stored in Ziploc plastic bags.

### 4.2. Isolation of Chitin Scaffolds

Samples of dried *A. aerophoba* sponge were carefully cut to produce square-shaped scaffolds with sizes of about 2 × 2 × 2 cm. The isolation of chitin scaffolds was accomplished using the previously reported method based on NaOH treatment [30,64]. The isolation procedure was repeated three times to obtain colorless scaffolds.

### 4.3. Dissolution of A. aerophoba Chitin in LiOH

*A. aerophoba* sponges were purified from other marine (not sponge) material using scissors. They were put into 500 mL of distilled water and sonicated for 1 h at room temperature. The sponges were dried using paper and inserted into ca. 500 mL of 20% acetic acid, after that, the content was sonicated for 2 h at 40 °C. The acetic acid solution was changed for a fresh portion and the samples were sonicated for another 2 h. The sponges were rinsed in distilled water till pH = 6.5. After that, the sponges were put into 500 mL of 1% LiOH (Sigma Aldrich, St. Louis, MO, USA) solution and sonicated at 50–55 °C for 50 h. The obtained scaffolds were rinsed in water and put for ca. 20 h in 20% acetic acid to remove residual calcium carbonate debris. The scaffolds were put into fresh 100 mL 1% LiOH solution and sonicated for 15 h at up to 65 °C and changed into the 1% LiOH solution for a fresh portion every day (ca. 100 mL). Next, the solution was dialyzed using membrane with the molecular weight cut-off 14 kDa (Carl Roth GmbH, Karlsruhe, Germany) in distilled water for 48 h at room temperature by changing the water every 2 h and checking the pH. Finally, the solution was lyophilized with Alpha 1–2 LDplus freeze dryer (Martin Christ GmbH, Osterode am Harz, Germany) during 24 h at −35 °C. The scheme of the procedure used in the study is presented in the Figure 12.

### 4.4. Chemicals

Lithium hydroxide monohydrate (pure p.a.) and glacial acetic acid (pure p.a.) were purchased from Chempur (Piekary Śląskie, Poland). The chitosan standard with molecular weight ~200,000 and deacetylation degree ≥ 90% was purchased from Pol-Aura (Poznań, Poland). Snow crab chitin was obtained from INTIB GmbH (Freiberg, Germany). Amorphous chitin was obtained from Prof. Rangzami Yajakumar (Amrita University, Bengaluru, Karnataka, India).

### 4.5. FTIR Spectroscopy

Fourier transformed infrared spectra of all the samples were recorded with the Nicolet iS50 FTIR spectrometer (Thermo Scientific, Inc., Waltham, MA, USA). Each analysis was performed using a built-in attenuated total reflectance (ATR) accessory. The measurements were carried out in the wavelength range of 4000–400 cm^−1^. Postprocessing of the recorded spectra was performed with OriginLab 2023 (OriginPro, Version 2023. OriginLab Corporation, Northampton, MA, USA).

### 4.6. X-ray Diffraction

The X-ray diffraction analysis was performed with the powder diffractometer SmartLab Rigaku, Japan with a CuK alpha lamp, in the 2θ range of 3–80°, scan step 0.01, scan speed 4°/min.

### 4.7. Digital Microscopy

The organic-free samples of the pure chitin scaffolds, partially dissolved scaffolds, and the film of lyophilized dissolved chitin from *A. aerophoba* were observed with the advanced imaging systems consisting of a VHX-6000 digital microscope (Keyence, Osaka, Japan) and VH-Z20R zoom lenses (magnification up to 200×) as well as Keyence VHX-7000 digital optical microscope with zoom lenses VHX E20 (magnification up to 100×) and VHX E100 (magnification up to 500×) (Keyence, Osaka, Japan).

### 4.8. Scanning Electron Microscopy (SEM) with Energy Dispersive X-ray Analysis (EDX)

The analyses were performed using the scanning electron microscope Quanta 250 FEG (FEI Ltd., Brno, Czech Republic) correlated with an energy dispersive X-ray spectrometer EDX Team Software. Additionally, selected samples were analyzed using SEM images with a scanning electron microscope (XL 30 ESEM, Philips, Amsterdam, The Netherlands). Prior to scanning, the samples were coated with a gold layer using the Cressington Sputtercoater 108 auto, Crawley (GB) (sputtering time 45 s).

### 4.9. Calcofluor White (CFW) Staining

CFW staining was used for the identification of β-(1-3) and β-(1-4) linked polysaccharides, including chitin. Samples were placed into a few drops of 0.1 M KOH-glycerine-water solution (solution A) and then a few drops of 0.1% CFW solution (Fluorescent Brightener M2R, Sigma-Aldrich, St. Louis, MO, USA) were added. The samples were placed in the dark place for 24 h. Afterwards, samples were rinsed three times with distilled water and dried at room temperature.

### 4.10. Fluorescent Microscopy

For fluorescence microscopy, the digital fluorescence microscope Keyence BZ9000 (Keyence, Osaka, Japan) was used. Fluorescence microscopy images were obtained using zoom lenses CFI Plan Apo 10×, CFI Plan Apo 40× through the DAPI channel (Ex/Em = 360/460 nm). The bright field regime was used for comparison.

## Figures and Tables

**Figure 1 marinedrugs-21-00334-f001:**
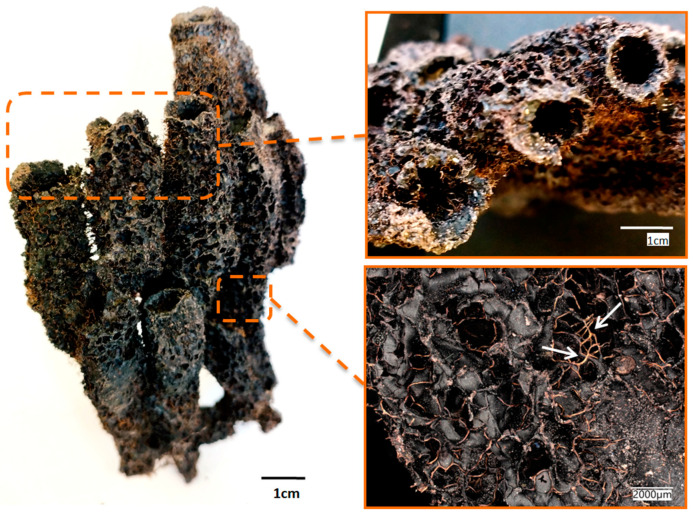
Image of the dried fragment of the *A. aerophoba* sponge with finger-like bioarchitecture. The chitin-based skeletal microfibers became visible (arrows) during the drying of the sponge body.

**Figure 2 marinedrugs-21-00334-f002:**
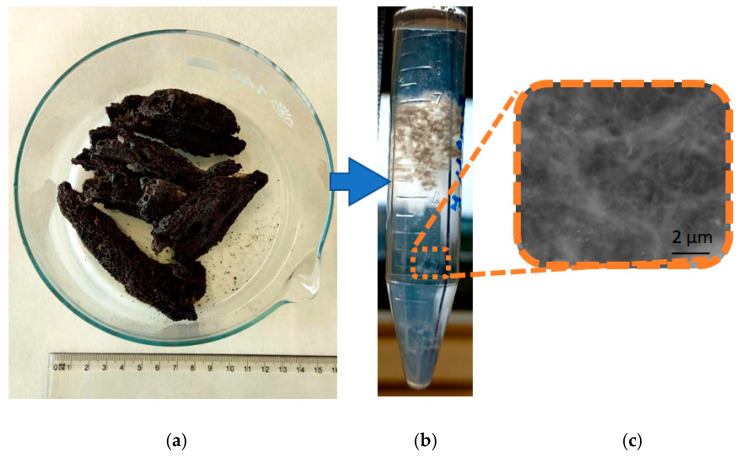
The sample of *A. aerophoba* sponge before the experiment (**a**) (see also Figure 1) and after insertion into 1% LiOH solution when disintegrated chitin scaffolds have been obtained (**b**). SEM image (**c**) shows non-regular microfibrous matter without typical microarchitecture observed with chitin scaffolds.

**Figure 3 marinedrugs-21-00334-f003:**
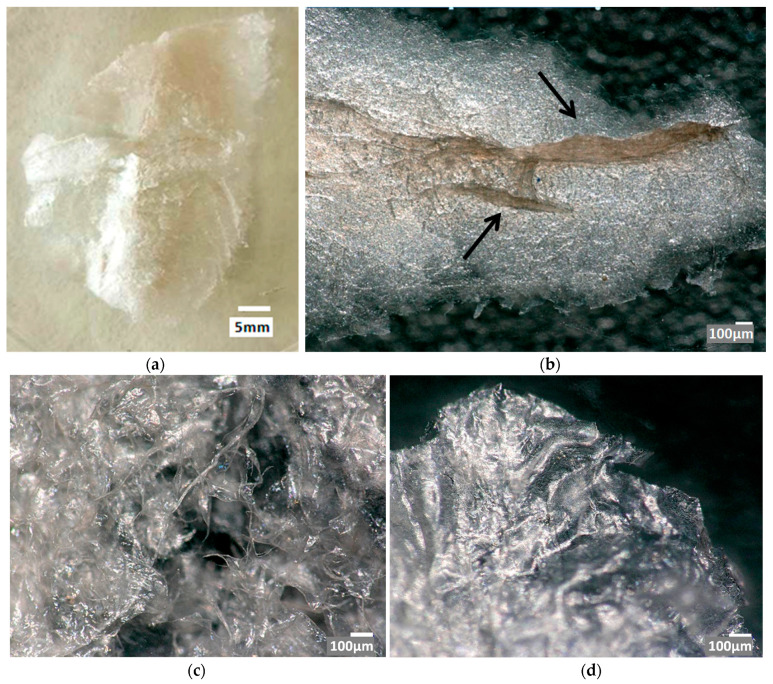
Dissolved and next dialyzed and lyophilized chitin of *A. aerophoba* sponge (**a**) after LiOH treatment. Digital microscopy images represent the partially dissolved chitin scaffold (**b**), where a few residual, pigmented chitinous microfibers (arrows) are still visible, and the finally-formed film (**c**,**d**).

**Figure 4 marinedrugs-21-00334-f004:**
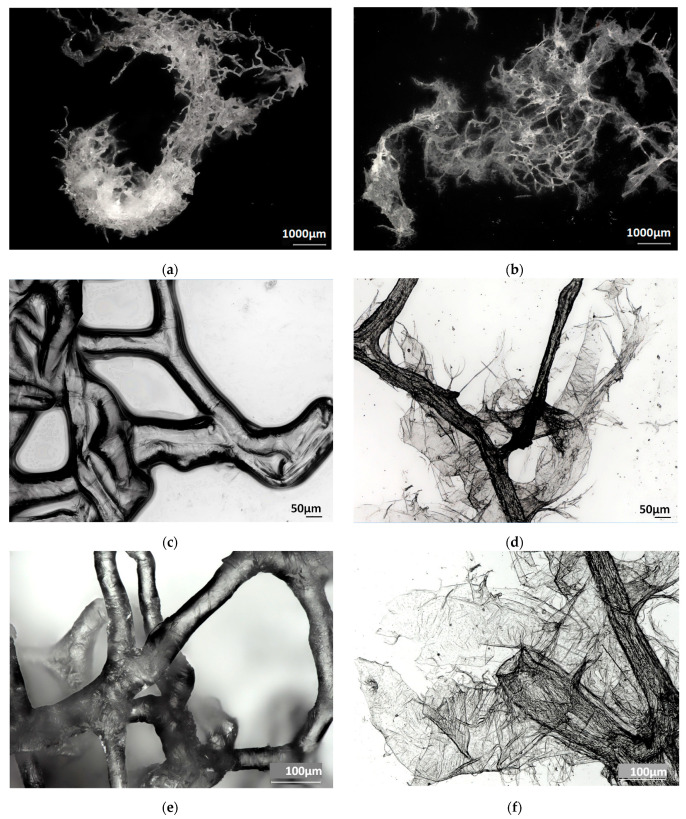
Digital microscopy imagery of isolated chitin scaffolds of *A. aerophoba* demosponge origin after 10% NaOH treatment (**a**,**c**,**e**) vs. chitin fibers partially dissolved in 1% LiOH (**b**,**d**,**f**).

**Figure 5 marinedrugs-21-00334-f005:**
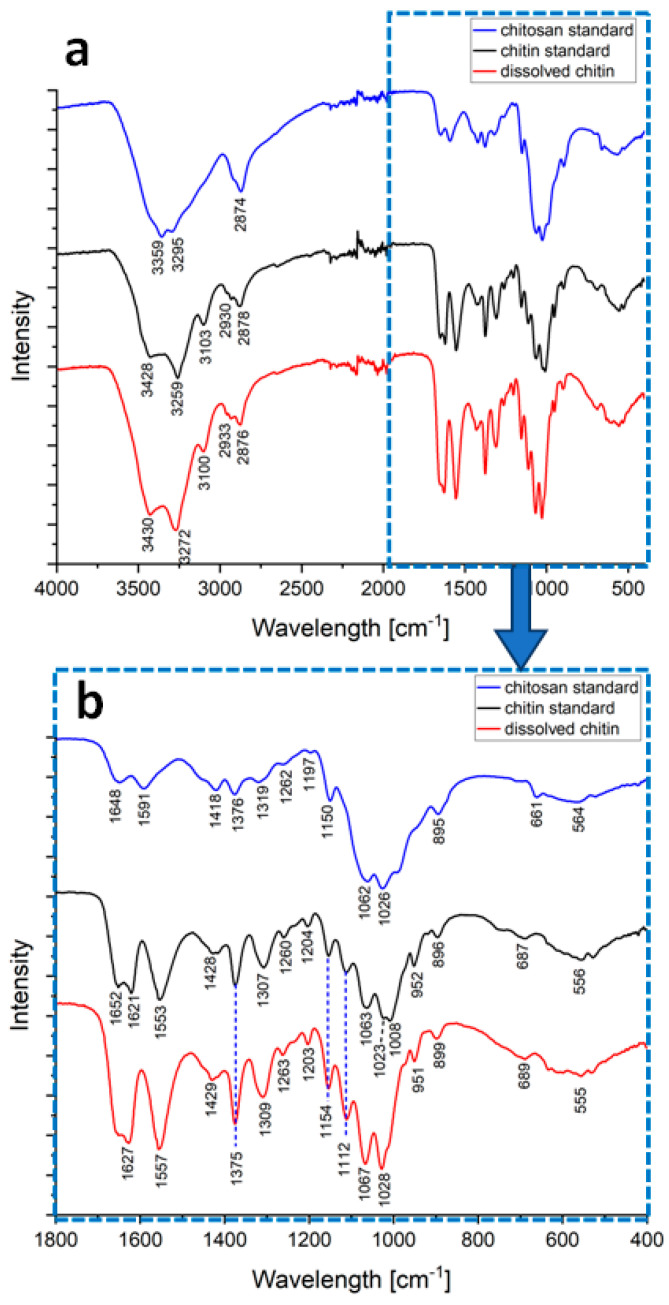
FTIR spectra of chitin and chitosan standards and in LiOH-dissolved sponge chitin in the range (**a**) 400—4000 cm^−1^ and (**b**) 400—1800 cm^−1^.

**Figure 6 marinedrugs-21-00334-f006:**
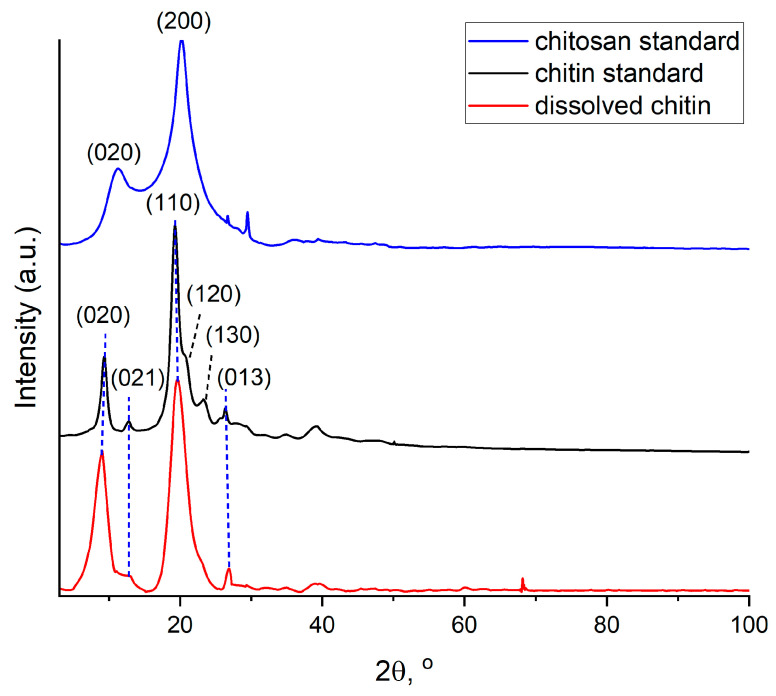
Normalized XRD spectra of chitin and chitosan standards in comparison to dissolved *A. aerophoba* sponge chitin.

**Figure 7 marinedrugs-21-00334-f007:**
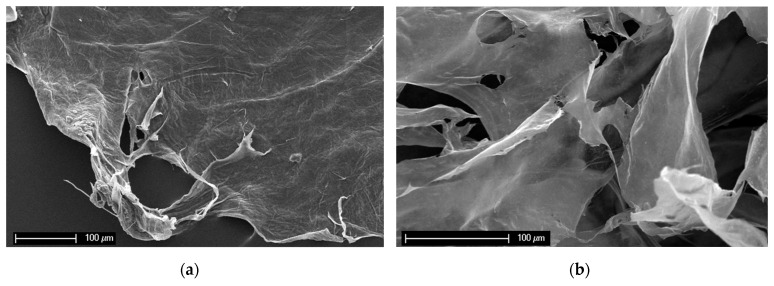
SEM images of the *A. aerophoba* chitin scaffold after NaOH treatment (**a**,**c**) and after dissolution in LiOH (**b**,**d**). The destruction of the structural integrity after insertion into LiOH solution on the microlevel is clearly visible.

**Figure 8 marinedrugs-21-00334-f008:**
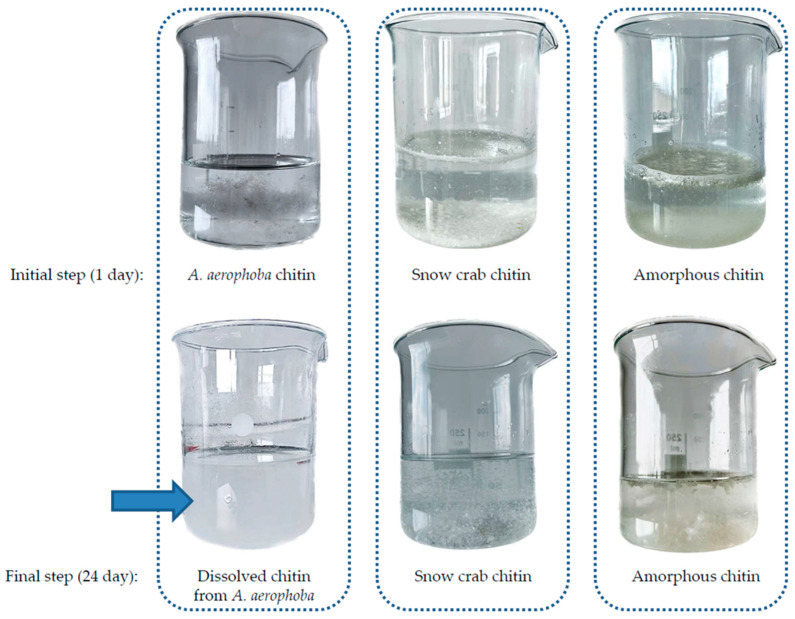
Comparison of different chitin samples before and after the LiOH-procedure. Only in the case of *A.aerophoba* sponge chitin was the milky suspension (arrow) obtained. The structural peculiarities of this phase are represented in the Figure 7d.

**Figure 9 marinedrugs-21-00334-f009:**
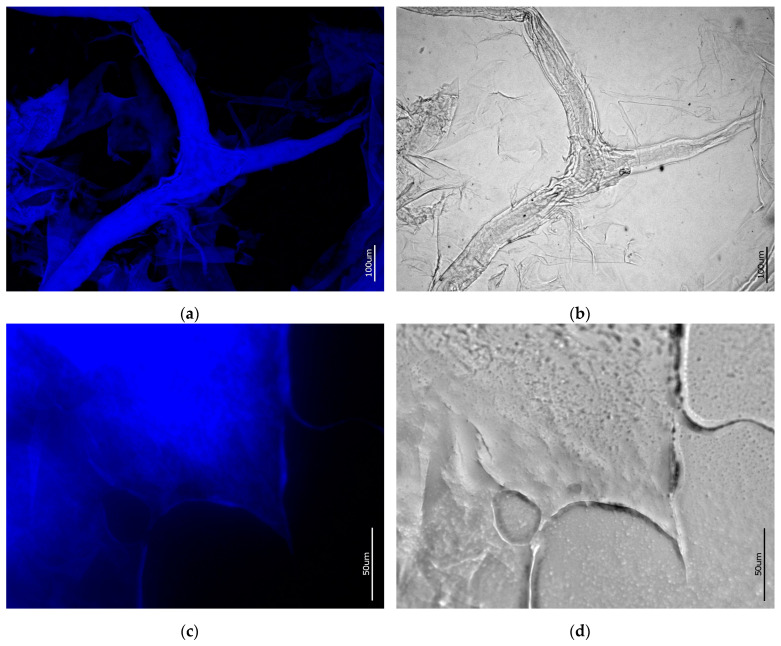
CFW-stained samples of (**a**,**b**) partially dissolved chitin fibers; (**c**,**d**) dried film of the obtained lyophylisate. Images (**a**,**c**) were obtained via the DAPI channel, images (**b**,**d**)—using bright field conditions. Light exposure time: (**a**) 1/6800 s; (**c**) 1/1100 s.

**Figure 10 marinedrugs-21-00334-f010:**
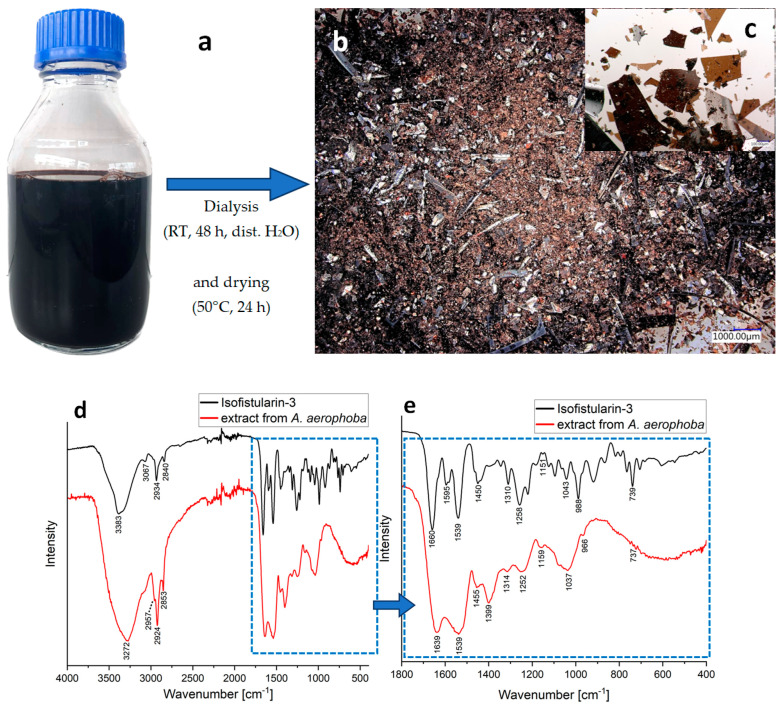
Bromotyrosine-containing extract (**a**) obtained during the chitin dissolution procedure based on LiOH treatment. (**b**) Digital microscopy image of the dialyzed and dried extract, (**c**) image in higher magnification. (**d**) Comparative FTIR spectra of the obtained dialyzed and dried extract and the Isofistularin-3 standard in the range 400–4000 cm^−1^ and (**e**) in the range 400–1800 cm^−1^.

**Figure 12 marinedrugs-21-00334-f012:**
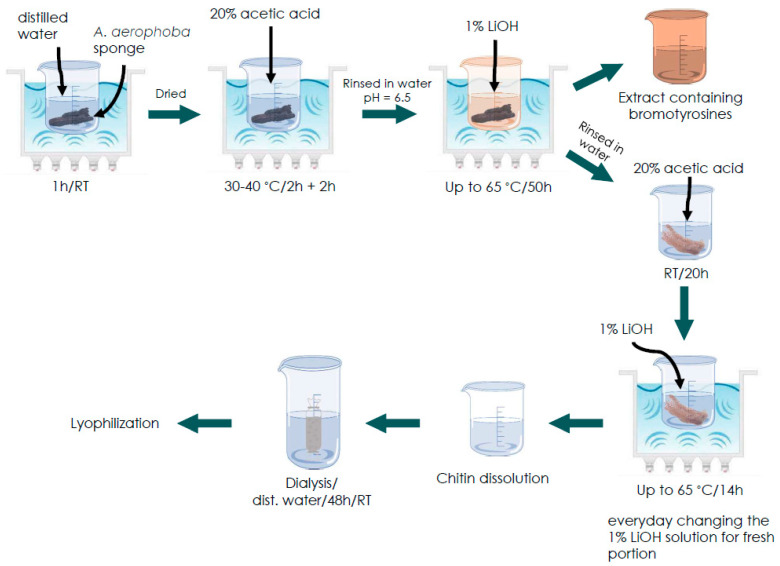
Schematic presentation of the *A. aerophoba* sponge chitin dissolution procedure in LiOH.

**Table 2 marinedrugs-21-00334-t002:** Wavenumbers from the FTIR spectra of chitin and chitosan standards and dissolved chitin and its assignments [6,18,67,68,69].

Chitosan Standard (cm^−1^)	Chitin Standard (cm^−1^)	Dissolved Chitin (cm^−1^)	Peak Assignment
3359	3428	3430	O–H stretching
3295	3259	3272	N–H stretching
-	3103	3100	N–H stretching
-	2930	2933	CH*_x_* stretching
2874	2878	2876	CH*_x_* stretching
1648	1652	-	Amide I
-	1621	1627	Amide I
1591	1553	1557	Amide II
1418	1428	1429	CH_2_ bending
1376	1375	1375	CH_3_ deformation
1320	1308	1309	Amide III
1262	1260	1263	Amide III
1197	1204	1203	Amide III
1150	1154	1154	C–O–C, C–O stretching
-	1112	1112	C–O–C, C–O stretching
1062	1063	1067	C–O–C, C–O stretching
1026	1023	1028	C–O–C, C–O stretching
-	1008	-	C–O stretch in phase ring
-	952	951	CH_3_ wagging
895	896	899	CH ring stretching

## Data Availability

The original data presented in the study are included in the article; further inquiries can be directed to the corresponding author.

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
