# Peer review of "The Loss of Structural Integrity of 3D Chitin Scaffolds from Aplysina aerophoba Marine Demosponge after Treatment with LiOH"

_marinedrugs, 2023, doi:10.3390/md21060334_

Round 1

Reviewer 1 Report

The manuscript describes the use of 1% LiOH solution as the solvent together with sonication for chitin extraction and dissolution from Aplysina aerophoba marine demosponge. The obtained chitin samples were characterized, and the different types of sources were attempted. Overall, it can be published after necessary revisions:

1.     It would be more interesting if the different types of common bases could be tried and revealed information on the influence of base species.

2.     The extraction process can be optimized, can more dilute base be used? Or the treatment capacity can be improved or not? (using less reagent to process more amount of substrate).

3.     The current characterization of chitin sample is not very comprehensive. For example, XPS can be used to see the valence change, BET can be used to see the porosity before and after treatment.

4.     There is some format error for Figure 12.

5.     Recently, shell biorefinery which is to utilize chitin as a platform resource to produce valuable organonitrogen chemicals have been proposed and emerged with rapid progress. This new relevant advance could be mentioned in the Introduction.

The English writing is good but still some minor improvements can be made by carefully examining the manuscript.

Author Response

The manuscript describes the use of 1% LiOH solution as the solvent together with sonication for chitin extraction and dissolution from Aplysina aerophoba marine demosponge. The obtained chitin samples were characterized, and the different types of sources were attempted. Overall, it can be published after necessary revisions:

Thank you for your time spent on the manuscript and for all the comments.

  1. It would be more interesting if the different types of common bases could be tried and revealed information on the influence of base species.

We have included a comprehensive table (Table 1) outlining the advantages and disadvantages of the different types of solvents for the dissolution of chitin published in the literature, including LiI, LiCl, NaOH, KOH etc. Your kind suggestion to use different types of alkali solutions could be explored in future comparative studies. Thank you for your valuable feedback.

  1. The extraction process can be optimized, can more dilute base be used? Or the treatment capacity can be improved or not? (using less reagent to process more amount of substrate).

We would like to emphasise that this is a preliminary “Proof-of-Concept” study to evaluate if a minimal 1% LiOH concentration practically suitable for laboratory work can yield the desired results/maintain scaffold integrity. However, for industrial/scaling-up work or further studies experimental protocols can be fine-tuned and modified to align with industrial processes. Future studies can further built on the results we have presented in our study. We would like to kindly suggest that even if the concentration of the alkali is reduced by half, the results would remain similar. Time is also a variable in our study, so reducing the time in the sonication bath can be fine-tuned for scaling-up work and future studies. This is the first study using LiOH for the isolation of sponge chitin.

  1. The current characterization of chitin sample is not very comprehensive. For example, XPS can be used to see the valence change, BET can be used to see the porosity before and after treatment.

Thank you for your comment. We have noted additional characterisation requirements and see them fit for future studies. As this is the first “Proof-of-Concept” study we would like to kindly emphasise that the focus of this study was on the structural integrity of the isolated chitin/development of new chemical process. We take your suggestions suitable for more applied studies in the future when the method is optimised. We kindly note that the porosity and other technical data about the scaffolds is already published and available in studies on other sponge species, such as:

Wysokowski et al. 3D Chitin Scaffolds of Marine Demosponge Origin for Biomimetic Mollusk Hemolymph-Associated Biomineralization Ex-Vivo. Mar Drugs. 2020 Feb 19;18(2):123. doi: 10.3390/md18020123.

Properties such as porosity can be further controlled by sonication and chemical pre-treatment.

  1. There is some format error for Figure 12.

Thank you for this comment, the error has been corrected.

  1. Recently, shell biorefinery which is to utilize chitin as a platform resource to produce valuable organonitrogen chemicals have been proposed and emerged with rapid progress. This new relevant advance could be mentioned in the Introduction.

Thank you, shell biorefinery is very interesting way of utilization chitin. It has been added to the revised article.

Reviewer 2 Report

The work by Izabela Dziedzic, Alona Voronkina, Martyna Pajewska-Szmyt, Martyna Kotula, Anita Kubiak, Heike Meissner, Tomas Duminis and Hermann Ehrlich is a very interesting work on the evolution of the chitin skeleton of Aplysina aerophoba when subjected to LiOH.

The work is in line with other reports on marine sponges with chitin structures. It present valuable information but in my opinion several interpretations of the results may be leading to a different conclusion that the ones obtained by the authors. In my comments, I will present the reasons why I came to different conclusions than the authors. I think they should be taken into account before publication. If the authors think different from me, please let me know your reason, so I can understand in a clear way your work.

Specific comments:

1-Figure 5: When comparing chitin and chitosan spectra, the authors could compare the acetylation degree of the chitins so to confirm the range of acetylation in the typical chitin acetylation degree range.

2-Figure 5: The loss of sharpness on the Amide I doublet supports authors conclusions regarding the dissolution of chitin since the presence of this doublet is associated to the crystalline structure of chitin.

3-Figure 6: In order to do a proper comparison of the XRD patterns, the dissolved chitin pattern should be normalized with the chitin standard pattern.

4-Figure 6: If dissolution takes places, amorphization should also occur. If this is correct, crysitallinity % should drop and this could be approximated by using the Segall approximation (similar to the one used for cellulose). It is not an accurate method, but could indicate the crystallinity variation among samples.

5-Section 2.7: The authors assigned the amide and C-O bands from the extract spectra to the moieties of the Bromotyrosine molecules. Since the authors also claim that chitin is solubilized and also presents these moieties, how can authors be sure that these bands became only from bromotyrosine and not the spectral sum of bromotyrosine and chitin?

6-Line 261: "Nothing is known about the possible role of bromotyrosines as cross-linking agents between the skeletal nanofibers...." Is there any evidence that bromotyrosines are cross-linking agents?

7-Line 293: " Upon heating the LiBr solution, cellulose undergoes complete dissolution" Is there any report that LiBr dissolves chitin or do the autors mean systems of LiX/solvent (DMAC, MeOH, etc)?

Author Response

The work by Izabela Dziedzic, Alona Voronkina, Martyna Pajewska-Szmyt, Martyna Kotula, Anita Kubiak, Heike Meissner, Tomas Duminis and Hermann Ehrlich is a very interesting work on the evolution of the chitin skeleton of Aplysina aerophoba when subjected to LiOH.

The work is in line with other reports on marine sponges with chitin structures. It present valuable information but in my opinion several interpretations of the results may be leading to a different conclusion that the ones obtained by the authors. In my comments, I will present the reasons why I came to different conclusions than the authors. I think they should be taken into account before publication. If the authors think different from me, please let me know your reason, so I can understand in a clear way your work.

Specific comments:

1-Figure 5: When comparing chitin and chitosan spectra, the authors could compare the acetylation degree of the chitins so to confirm the range of acetylation in the typical chitin acetylation degree range.

Thank you for your time spent on the manuscript and for all the comments. The aim of the comparison was showing that in the conditions in the manuscript chitin does not transform to chitosan. A comparison of the degree of acetylation between the different samples could be made in future studies.

2-Figure 5: The loss of sharpness on the Amide I doublet supports authors conclusions regarding the dissolution of chitin since the presence of this doublet is associated to the crystalline structure of chitin.

Thank you very much. We have added this suggestion to the revised manuscript.

3-Figure 6: In order to do a proper comparison of the XRD patterns, the dissolved chitin pattern should be normalized with the chitin standard pattern.

Thank you for your valuable comment. We have normalised the spectra now and included it in the revised manuscript.

4-Figure 6: If dissolution takes places, amorphization should also occur. If this is correct, crystallinity % should drop and this could be approximated by using the Segall approximation (similar to the one used for cellulose). It is not an accurate method, but could indicate the crystallinity variation among samples.

Thank you for your suggestion. We agree with the reviewer that some degree of amorphization will occur, however, form Figure 4 it can be observed that the fibres have lost some structural integrity but are still intact. So the degree of amorphization from Segall approximation is probably less than 5% and could be further evaluated in different studies assessing concentration of alkali and degree of amorphization.

5-Section 2.7: The authors assigned the amide and C-O bands from the extract spectra to the moieties of the Bromotyrosine molecules. Since the authors also claim that chitin is solubilized and also presents these moieties, how can authors be sure that these bands became only from bromotyrosine and not the spectral sum of bromotyrosine and chitin?

Thank you for your comment. We would like to kindly note that bromotyrosine extract preparation is “separated” from the sponges (Figure 12) and collected before dissolution of chitin scaffold.

6-Line 261: "Nothing is known about the possible role of bromotyrosines as cross-linking agents between the skeletal nanofibers...." Is there any evidence that bromotyrosines are cross-linking agents?

Yes, previously 3,5-dibromotyrosines have been reported as cross-linking agents in cuticle of Atlantic horseshoe crab (Limulus polyphemus) (Welinder, 1972) as well as within an operculum scleroprotein from the large whelk Buccinum undatum (Hunt & Breuer, 1973). We made corresponding changes in the revised manuscript.

Welinder, B.S. Halogenated Tyrosines from the Cuticle of Limulus Polyphemus (L.). Biochim Biophys Acta Gen Subj 1972, 279, 491–497, doi:10.1016/0304-4165(72)90171-7.

Hunt, S.; Breuer, S.W. Chlorinated and Brominated Tyrosine Residues in Molluscan Scleroprotein. Biochem. Soc. Trans. 1973, 1, 215–216, doi:10.1042/bst0010215.

7-Line 293: " Upon heating the LiBr solution, cellulose undergoes complete dissolution" Is there any report that LiBr dissolves chitin or do the autors mean systems of LiX/solvent (DMAC, MeOH, etc)?

Thank you for your kind comment. Our work is first to dissolve and isolate chitin scaffold in LiBr, other biopolymers such as cellulose can be dissolved completely in LiBr solution. We have inserted corresponding references concerning the dissolution of chitin in LiBr under diverse, selected conditions reported by other authors into the revised manuscript.

For example such as:

Gözaydın, G.; Song, S.; Yan, N. Chitin Hydrolysis in Acidified Molten Salt Hydrates. Green Chem. 2020, 22, 5096–5104, doi:10.1039/D0GC01464H.

Gözaydın, G.; Sun, Q.; Oh, M.; Lee, S.; Choi, M.; Liu, Y.; Yan, N. Chitin Hydrolysis Using Zeolites in Lithium Bromide Molten Salt Hydrate. ACS Sustainable Chem. Eng. 2023, 11, 2511–2519, doi:10.1021/acssuschemeng.2c06675.

We have also improved the manuscript and incorporated the changes. Thank you very much.

Round 2

Reviewer 1 Report

The comments have been addressed and can be published.

Good